# Mathematical Modelling of Virus Spreading in COVID-19

**DOI:** 10.3390/v15091788

**Published:** 2023-08-23

**Authors:** Liaofu Luo, Jun Lv

**Affiliations:** 1Faculty of Physical Science and Technology, Inner Mongolia University, 235 West College Road, Hohhot 010021, China; 2College of Science, Inner Mongolia University of Technology, 49 Aymin Street, Hohhot 010051, China

**Keywords:** virus spread, COVID-19, mathematical model, daily number of infections, cumulative number of infections

## Abstract

A mathematical model is proposed to analyze the spreading dynamics of COVID-19. By using the parameters of the model, namely the basic reproduction number (*R*_0_) and the attenuation constant (*k*), the daily number of infections (DNI) and the cumulative number of infections (CNI) over time (*m*) are deduced and shown to be in good agreement with experimental data. This model effectively addresses three key issues: (1) inferring the conditions under which virus infections die out for a specific strain given *R*_0_; (2) explaining the occurrence of second waves of infection and developing preventive measures; and (3) understanding the competitive spread of two viruses within a region and devising control strategies. The findings highlight the potential of this simple mathematical framework in comprehensively addressing these challenges. The theoretical insights derived from this model can guide the evaluation of infection wave severity and the formulation of effective strategies for controlling and mitigating epidemic outbreaks.

## 1. Introduction

The ongoing COVID-19 pandemic has posed several unresolved questions. Firstly, our aim is to understand the reason behind the waveform pattern observed in the daily number of infections (DNI) and to predict the scale and duration of an infection wave, including the cumulative number of infections (CNI) and the time until infection ends. Secondly, we investigate why the second wave of infection typically follows the first wave and how to predict its occurrence. Lastly, we explore how virus spreading in a region is influenced by the emergence of new strains and develop methods to control their spread. Existing epidemic models have limitations in addressing these problems comprehensively. Since the 1920s, differential equations were used to model the population distribution of disease spread, including susceptible, infected, and recovered/dead pools [1,2]. Such models could not examine important social and behavioural factors, such as the behavioural responses of individuals to policy measures, and the effect of heterogeneous social contacts on diffusion patterns. Next, since the 1990s, agent-based simulations were proposed that included some important sources of population heterogeneity and explored the structure and dynamics of transmission networks. However, none of the agent-based models are based on explicit empirical/theoretical assumptions of individual behaviour, social transmission mechanisms, and social structure constraints [3,4,5]. Recently, several network models simulating the spread of epidemics in the population were proposed, such as the Susceptible-Infectious-Susceptible model on random networks [6], and the epidemic spreading on modular networks [7]. Although these specific models have solved some aspects of the complexity of infectious diseases and obtained meaningful results, they still fail to comprehensively answer the aforementioned unresolved questions.

As the Chinese idiom goes, “the greatest truths are the simplest”. Starting from the principle of natural selection—the interaction and compromise between the virus and its host (human)—we propose a mathematical model based on insights from experimental data. The model is based on two fundamental assumptions. First, each viral strain has a basic reproduction number (*R*_0_), which represents the average number of infections caused by an initial infectious person in a completely susceptible population [8]. *R*_0_ is a measure of the transmission potential of a particular infectious disease. Second, virus infections undergo a series of events due to the presence of population immunity or intervention that makes the instantaneous reproduction number, *R_t_*, decreasing step by step, described by an attenuation parameter *k* (*k* < 1) [9,10]. The parameter *k* is influenced by social contacts and policy measures, where looser contacts and measures result in higher values of *k*. It is worth noting that public health measures in a given region usually undergo significant changes over a longer period of time compared to the duration of a virus wave [11]. The stringency index of policy measures can be found at https://ourworldindata.org/covid-stringency-index (accessed on 13 August 2023). Therefore, it is reasonable to assume that the attenuation parameter *k* is approximately constant during a single wave of virus spread. Based on two parameters, *R*_0_ and *k*, we can deduce the general formula for the daily number of infections (DNI) and the cumulative number of infections (CNI) over *m* steps. The CNI is denoted by *F*(*R*_0_, *k*; *m*). These formulas are in good agreement with experiments and provide a generalized framework to simulate single and multiple virus infections. By introducing *F*_1_ and *F*_2_ we can also study the competitive spread of two viruses in a region, including scenarios with the introduction of a new virus strain. It is worth noting that the loosening of public health measures and/or the emergence of new viruses can lead to subsequent waves of virus transmission.

## 2. Materials and Methods

The data of daily virus infections of COVID-19 were taken from WHO (https://covid19.who.int/data, accessed on 12 May 2023) for the UK and from the public database (https://ourworldindata.org/coronavirus, accessed on 12 May 2023) for Hong Kong. Since Gauss discovered the least square method and successfully applied it to astronomical observation, the ordinary least square is recognized as one of the best methods for curvilinear regression. In the present study, we used the least square simulation of observational data of the virus infection to determine the parameters in each COVID-19 pandemic. The goodness of the nonlinear least square fitting (NLLSF) is tested by R^2^, i.e.,
$$R^{2}=1-\frac{\sum_{i=1}^{n}{\left(y_{i}-{\hat{\pi }}_{i}\right)}^{2}}{\sum_{i=1}^{n}{\left(y_{i}-\bar{y}\right)}^{2}},$$
where *y_i_* are the observation values, $\bar{y}$ their average, and ${\hat{\pi }}_{i}$ the predictions of the model. The goodness R^2^ and the *p*-value of Prob(F-statistic) are calculated for each simulation. The F-statistic is defined as
$$F\text{-}statistic=\frac{(\sum_{i=1}^{n}{\left(y_{i}-\bar{y}\right)}^{2}-\sum_{i=1}^{n}{\left(y_{i}-{\hat{\pi }}_{i}\right)}^{2})/(k-1)}{\sum_{i=1}^{n}\frac{{\left(y_{i}-{\hat{\pi }}_{i}\right)}^{2}}{n-k}},$$
(*n*–number of samples, *ν*–number of parameters) that can be calculated from the experimental data. The *p*-value is decided by the percentile of the Fisher’s F-distribution F*_ν_*_−1,*n*−*ν*_(*z*), namely,
$$p\text{-}value=the\;probability\;of\;F_{\nu -1,n-\nu }(z>F\text{-}statistic)$$

The goodness R^2^, the *p*-value, and the root mean squared error (RMSE) for each simulation are given in the figures.

## 3. Results

### 3.1. Derivation of Formulas for Cumulative Number of Infections (CNI) and Daily Number of Infections (DNI)

The potential for infection of a given virus strain is determined by the basic reproduction number known as *R*_0_. For the virus to spread, it must continually re-infect the rest of the susceptible population. However, countermeasures by humans must aim to limit these subsequent infections. Essentially, this implies that the interaction and compromise between the virus and host (humans) necessitates the *j*-th reproduction number *R_j_* to be smaller than the (*j* − 1)-th reproduction *R_j_*_−1_, as expressed in the equation
$$R_{j}=kR_{j-1},(k<1).$$

The reason *R_j_* changes with *j* is twofold. Firstly, as the virus continues to spread, the susceptible population decreases, leading to a decrease in the transmission potential. Secondly, ongoing public health measures are being implemented to control the spread of the virus, further contributing to the reduction in the instantaneous reproduction number. The attenuation constant ‘*k*’ in this scenario is influenced by social contact rates and policy measures. After *m* rounds of infection spread, the total number of infections in a specific area can be represented as
(1)$$F(R_{0},k;m)=R_{0}+R_{0}R_{1}+\cdots +R_{0}R_{1}R_{2}\cdots R_{m-1}=R_{0}+kR_{0}^{2}+\cdots +k^{m(m-1)/2}R_{0}^{m}$$

This function *F*(*R*_0_, *k*; *m*) describes the cumulative number of infections (CNI), which is an increasing function of *m*. Assuming the total number of infected individuals to be *N*, solving the equation *F*(*R*_0_, *k*; *m*) = *N* will produce ‘*m*’, the transmission number of the virus strain. The parameter *m* grows with time *t*. Assuming each transmission takes place every *q* days, we have
(2)$$m=(t+t_{0})/q$$
where *t*_0_ is a time shift parameter linked to the start of infection (the time of *m* = 1). Equations (1) and (2) provide the cumulative number of infections (CNI) for each branch of the epidemic. If various generations of different branches exist simultaneously, the total CNI is simply the sum of the CNIs of each branch. The CNI can be observed experimentally. Inserting (2) into (1), we obtain a formula of CNI vs. *t* represented by four parameters *R*_0_, *k*, *q*, and *t*_0_ that can be used to simulate the change of a cumulative number of infections with time in a region. As seen in Figure 1, our simulations of the UK-alpha strain (November 2020 to April 2021) and the Hong Kong-delta/omicron strain (February 2022 to April 2022) spreading are well fitted to the data obtained from COVID-19 pandemic updates. The high accuracy of the simulation shows that the assumption of the constancy of *k* is reasonable. It is important to note that our model not only allows for simulation of COVID-19 pandemics but also enables predictions of virus spread over longer periods by utilizing parameter data derived from a particular phase of the outbreak.

Applying Equations (1) and (2), the daily number of infections (DNI) is derived:(3)$$dF/dt=(dF/dm)/q,\;dF/dm=k^{m(m-1)/2}R_{0}^{m}.$$

From Equation (3), it is easy to understand why DNI first rises then falls, considering that *R*_0_ > 1 (for most infectious viruses) and *k* < 1. Zero DNI arrives when *dF*/*dm* = 0, which requires *m* to be large enough that
(4)$$k^{(m-1)/2}R_{0}<1\;or\;R_{0}<{\left(1/k\right)}^{(m-1)/2}$$

This is a condition for the end of an infection wave.

### 3.2. Insights from Typical Figures of CNI vs. m

Utilizing Equation (1), we plotted the change of CNIs with variables *k* and *m* for the given *R*_0_ (Figure 2). Figure 2A–E show typical cases that help in understanding the development and termination of an infection wave.

From Figure 2, one can observe that the curve of *F*(*R*_0_, *k*; *m*) increases with *m* and approaches a stable value when *m* > *m*_st_ for each *k* < *k*_th_. We denote *F*(*R*_0_, *k*_th_; *m*_st_) as *N*_th_. Calculating for *m*_st_ = 15 in Figure 2A–E, we obtained the threshold *k*_th_ and the corresponding CNI *N*_th_ for each given *R*_0_ (each virus strain). On the other hand, CNI attaining a stable value means the DNI is approaching zero. By defining *E*_km_ = (1/*k*_th_)^(*m*st−1)/2^, Equation (4) can be written as *R*_0_ < *E*_km_. The threshold *k*_th_, the corresponding CNI *N*_th,_ and the parameter *E*_km_ are listed in Table 1. We found the relation *R*_0_ < *E*_km_ to hold well across all data.

The daily increasing number *dF/dm* = 0 means the virus infection is dying out. The above result shows the higher the basic reproduction number *R*_0_, the stricter the public health measure is required to be to increase *E*_km_ to satisfy Equation (4) and terminate the wave of virus infection effectively. From Table 1, we found the cumulative infection numbers, *N*_th_, of many virus strains (except Omicron with *R*_0_ = 18.6) are lower than 10^5^ if an appropriate *k* (lower than *k*_th_) is introduced.

For the virus strain of given *R*_0_, if *F*(*R*_0_, *k*; *m*) (for all *m*) exceeds a threshold *N*_max_, then the spread of this strain would lead to a wave of COVID-19 infection in a region with a population larger than *N*_max_. In order to end this spread, the necessary condition would be
(5)$$F(R_{0},\;k;\;m)<N_{max}(for\;all\;m)$$

Equation (5) provides a constraint on *k* for strains with *R*_0_. The critical value of *k* is denoted as *k*_cr_. Taking *N*_max_ = 10^7^, the values of *k*_cr_ are also listed in Table 1. Note that the parameter *k*_th_ is the threshold value of *k* required for ending the spread in the 15th generation, but the parameter *k*_cr_ is that value for ending the spread in an arbitrary generation. The latter constraint is looser than the former. Therefore, *k*_cr_ is larger than *k*_th_.

Virus strains with low *R*_0_ values have higher *k*_cr_ and *k*_th_ values. As a result, they can spread in regions with smaller populations under looser public health measures. This theoretical prediction can explain why strains like SARS-CoV in 2003 only spread in restricted regions and soon disappeared globally. On the contrary, virus strains with high *R*_0_ values, such as Omicron, have lower *k*_cr_ and *k*_th_ values. To satisfy Equations (4) and (5), the parameter *k* needs to meet very strict constraints. In cases where social management measures are not stringent enough, the number of infected individuals will quickly surpass the *N*_max_ limit, triggering a global pandemic wave, possibly culminating in a type of coexistence between viruses and humans.

In summary, the termination of an infection wave is determined by the condition *dF/dm* = 0 on the CNI diagram (Figure 2), which requires a sufficiently large *m* and the condition in Equation (4) satisfied, i.e., *R*_0_ < *E*_km_ = (1/*k*_th_)^7^. Additionally, the necessary condition for a pandemic to die out in a population of *N*_max_ is expressed by Equation (5), imposing a limitation on *k*, namely *k* < *k*_cr_.

### 3.3. Prediction on the Second Wave of Pandemics

Early models based on previous pandemics such as SARS, MERS, and the 2009 H1N1 outbreak can effectively predict the occurrence of the first wave of a disease. However, their predictive power decreases when it comes to anticipating the possibility of a second wave [12]. This raises the question: why does the second wave of COVID-19 infections often follow the first wave? The present model aims to address this issue.

As shown in Figure 2, there are multiple curves (*F* versus *m*) for a given *R*_0_ value. These curves differ from each other by the parameter *k*. When the number of daily new infections (DNI) approaches zero, any fluctuation in *k* significantly influences the spread of the virus. Therefore, changes in public health measures can induce variations in virus spread along different curves. Generally, as public health measures are relaxed, the parameter *k* increases, causing the virus to transition from one curve of *F* to another steeper curve. This signifies the onset of a new wave of virus spread. For instance, in Figure 2A, the spread of the Omicron variant (with an *R*_0_ of 18.6) is plotted. Assuming the initial spread is along the curve with *k* equal to 0.613, a stable state is reached at *m* = 15. At this point, the *k* value increases to 0.722. In response to this change, the virus begins spreading along a new curve with *k* = 0.722, starting from *m* = 4, as the value of *F*(18.6, 0.722; *m* = 4) is equal to the original CNI value, *F*(18.6, 0.613; *m* = 15). This example explains how the second wave of virus infection occurs. However, in cases where the parameter *k* decreases when the DNI approaches zero, the curve of *F* will transition to a flatter one. This indicates that the first wave of viral infection will end soon, and no second wave will occur.

By examining Figure 2A–E, we can analyze how the occurrence of a second wave depends on the *R*_0_ value of the virus. For instance, as *k* changes from 0.85 to 0.9, the CNI (at *m* = 15) for high *R*_0_ viruses increases hundreds of times, whereas it only increases tens of times for low *R*_0_ viruses. Therefore, our model predicts that multiwave infections are more likely to occur with viruses that have high *R*_0_ values.

In the aforementioned discussions, we have assumed that no virus mutation occurs and that only one type of virus is spreading. In reality, changes in public health measures may be accompanied by virus mutations. In this case, the change in public health measures would cause the jump of *F* not only between different curves with a given *R*_0_ but also between different *R*_0_ values, providing more opportunities for the occurrence of a second wave during virus spread.

Another crucial point to consider is that the change in *k* can simultaneously induce a change in *q*, as the eigen-time (inherent time) *m* depends on *q* (Equation (2)). In the case of a single wave, the parameter *m* can be used to represent time dependence, and *q* can be simply set to 1 (known as normalization). However, when studying two continuous waves, the dependence on *q* should be clearly indicated due to the different eigen-times *m*. We have mentioned that the dependence of *F* on the change in *k* results in a jump from one curve to another. Meanwhile, the dependence of *F* on the change in *q* only affects the lengthening or shortening of the abscissa of the graph without altering the shape of the curve. When the public health measures change and a jump between curves occurs, the abscissa of the graph of the second wave simultaneously lengthens or shortens.

The occurrence of the second wave of infection is more likely when public health measures are relaxed. This change in the second wave is accompanied by a change in the *q* value. These predictions align with experimental data. For example, in the UK from May to September 2021, public health measures were relaxed, and a second wave of infection followed the first wave (Figure 1A). Similarly, in Hong Kong several months after May 2022, the looser public health measures led to a second wave occurring after the first wave (Figure 1B). (The data on the change of public health measures can be found at https://www.bsg.ox.ac.uk/research/covid-19-government-response-tracker, accessed on 13 August 2023).

In summary, the dependence of the CNI on *k* (given a specific *R*_0_) is determined by the jump between different curves on the graph *F*(*R*_0_, *k*; *m*) versus *m*, referred to as *k*-transformation. The dependence of the CNI on the duration of m is obtained by stretching the abscissa m on the graph, known as *q*-transformation. The *k*-transformation and *q*-transformation, occurring when the first wave is nearing its end, are the causes of a continuous second wave. The change in *k* is attributed to the modification of public health measures, while the modification of *q* is due to the change in physical time within a unit of *m*. The changes in *k* and *q* provide an explanation for the experimental data on continuous multiwave infections.

### 3.4. Cross-Spread of Two Viruses: Discriminant Function

Viral infections often involve the simultaneous presence of two or more viruses. To accurately simulate the cross-spread of two viruses, it is necessary to consider the differences in eigen-times (*m*_1_ and *m*_2_) between these viruses and their relationship with the physical time *t*. Consequently, additional parameters *q* and *t*_0_ (as given in Equation (2)) must be taken into account. By utilizing the four parameters *R*_0_, *k*, *q*, and *t*_0_, the CNI *F*(*R*_0_, *k*; *m*) can be expressed as:(6)$$F(R_{0},\;k;\;m(t))=F(t;\;a,\;b,\;c,\;d),\;(a=R_{0},\;b=k,\;c=q,\;d=t_{0})$$

When the CNIs of two virus strains *F*_1_(*t*; *a*_1_, *b*_1_, *c*_1_, *d*_1_) and *F*_2_(*t*; *a*_2_, *b*_2_, *c*_2_, *d*_2_) intersect at *t*_cr_,
(7)$$F_{1}(t_{cr};\;a_{1},\;b_{1},\;c_{1},\;d_{1})=F_{2}(t_{cr};\;a_{2},\;b_{2},\;c_{2},\;d_{2})$$
and
$$\begin{array}{c} F_{1}(t;\;a_{1},\;b_{1},\;c_{1},\;d_{1})>F_{2}(t;\;a_{2},\;b_{2},\;c_{2},\;d_{2})\;as\;t<t_{cr},\\ F_{1}(t;\;a_{1},\;b_{1},\;c_{1},\;d_{1})<F_{2}(t;\;a_{2},\;b_{2},\;c_{2},\;d_{2})\;as\;t>t_{cr}. \end{array}$$

This represents a transition in the population of virus strains from *F*_1_ to *F*_2_. As it is challenging to directly solve the intersection equation (Equation (7)), we introduce a function:(8)$$D_{21}=log[(dF_{2}/dt)/(dF_{1}/dt)]$$
which satisfies:(9)$$\begin{array}{c} D_{21}>0\;\mathrm{as}\;dF_{2}/dt>dF_{1}/dt,\\ D_{21}<0\;\mathrm{as}\;dF_{2}/dt<dF_{1}/dt. \end{array}$$

By utilizing Equations (2), (3), (6) and (8), *D*_21_ can be formulated as a simple quadratic function of time
(10)$$D_{21}=\alpha t^{2}+\beta t+\gamma$$
where
$$\begin{array}{c} \alpha =\log b_{2}/\left(2c_{2}^{2}\right)-\log b_{1}/\left(2c_{1}^{2}\right),\;\\ \beta =(2d_{2}-c_{2})\log b_{2}/(2c_{2}^{2})-(2d_{1}-c_{1})\log b_{1}/(2c_{1}^{2})+\log a_{2}/c_{2}-\log a_{1}/c_{1},\;\\ \gamma =d_{2}(d_{2}-c_{2})\log b_{2}/(2c_{2}^{2})-d_{1}(d_{1}-c_{1})\log b_{1}/(2c_{1}^{2})+d_{2}\log a_{2}/c_{2}-d_{1}\log a_{1}/c_{1}+\log(c_{1}/c_{2}). \end{array}$$

In order to determine if a real root of *D*_21_ exists, we define:(11)$$∆=\beta^{2}-4\alpha \gamma$$

The value of Δ determines the existence of the real root in the quadratic form *D*_21_. This form, known as the discriminant function, serves as a tool to identify the occurrence of *t*_cr_ and the domain of its existence. The prediction rules can be summarized as follows:

Rule 1: When Δ > 0, the quadratic form intersects with the *t* axis at *t_s_* and *t_m_* (*t_m_* > *t_s_*),
$$t_{s}=\left\{\begin{array}{c} \frac{-\beta -\sqrt{∆}}{2\alpha }\;(\mathrm{for}\;\alpha >0)\\ \frac{-\beta +\sqrt{∆}}{2\alpha }\;(\mathrm{for}\;\alpha <0) \end{array}\right.,\;t_{m}=\left\{\begin{array}{c} \frac{-\beta +\sqrt{∆}}{2\alpha }\;(\mathrm{for}\;\alpha >0)\\ \frac{-\beta -\sqrt{∆}}{2\alpha }\;(\mathrm{for}\;\alpha <0) \end{array}\right..$$

These values partition the time into three distinct domains: *t* < *t*_s_ in the first domain, *t*_s_ < *t* < *t*_m_ in the second domain, and *t* > *t*_m_ in the third domain.

Rule 2: In the first domain, *t*_a_ = *qm*_0_ − *t*_0_ is defined as the initial time where *m*_0_ >> 1, indicating *m*_1,2_ = (*t* + *d*_1,2_)/*c*_1,2_ >> 1. If *D*_21_ > 0, there will be no intersection of *F*_1_(*t*) and *F*_2_(*t*) in the domain between *t*_a_ and *t*_s_ when the initial values of *F* at *t*_a_ satisfy *F*_1_(*t*_a_) < *F*_2_(*t*_a_), but there may be one *t*_cr_ (the number of *t*_cr_ is either 1 or 0) in the domain when *F*_1_(*t*_a_) > *F*_2_(*t*_a_). If *D*_21_ < 0, there will be no intersection of *F*_1_(*t*) and *F*_2_(*t*) in the domain when the initial values satisfy *F*_1_(*t*_a_) > *F*_2_(*t*_a_), but there may be one *t*_cr_ (the number of *t*_cr_ is either 1 or 0) when *F*_1_(*t*_a_) < *F*_2_(*t*_a_).

Rule 3: In the second domain, if *D*_21_ > 0 there will be no intersection of *F*_1_(*t*) and *F*_2_(*t*) in the domain when the *F*-values at *t* = *t*_s_ satisfy *F*_1_(*t*_s_) < *F*_2_(*t*_s_), but there may be one *t*_cr_ (the number of *t*_cr_ is either 1 or 0) when *F*_1_(*t*_s_) > *F*_2_(*t*_s_). If *D*_21_ < 0 there will be no intersection of *F*_1_(*t*) and *F*_2_(*t*) in the domain when the *F*-values at *t* = *t*_s_ satisfy *F*_1_(*t*_s_) > *F*_2_(*t*_s_), but there may be one *t*_cr_ (i.e., the number of *t*_cr_ is either 1 or 0) when *F*_1_(*t*_s_) < *F*_2_(*t*_s_). The *F*-values at *t* = *t*_s_ are determined by *F*_1_(*t*) and *F*_2_(*t*) in the first domain.

Rule 4: In the third domain, if *D*_21_ > 0 there will be no intersection of *F*_1_(*t*) and *F*_2_(*t*) in the domain when the *F*-values at *t* = *t*_m_ satisfy *F*_1_(*t*_m_) < *F*_2_(*t*_m_), but there may be one *t*_cr_ (the number of *t*_cr_ is either 1 or 0) when *F*_1_(*t*_m_) > *F*_2_(*t*_m_). If *D*_21_ < 0 there will be no intersection of *F*_1_(*t*) and *F*_2_(*t*) in the domain when the *F*-values at *t* = *t*_m_ satisfy *F*_1_(*t*_m_) > *F*_2_(*t*_m_), but there may be one *t*_cr_ (i.e., the number of *t*_cr_ is either 1 or 0) when *F*_1_(*t*_m_) < *F*_2_(*t*_m_). The *F*-values at *t* = *t*_m_ are determined by *F*_1_(*t*) and *F*_2_(*t*) in the second domain.

Rule 5: There can be at most one *t*_cr_ in a given domain because the symbol of *D*_21_ is definite in any domain. The magnitude of the domain is an important factor to predict the occurrence of a *t*_cr_. For example, in the second domain the magnitude is *t*_m_* − t*_s_ = Δ^1/2^/(2|*α*|), and the necessary condition for the occurrence of *t*_cr_ is a large enough (*t*_m_ − *t*_s_) or Δ.

Rule 6: When Δ < 0, the quadratic form *D*_21_ does not intersect with *t* axis. In this case, there is only one domain, and the rule is the same as that in the first domain given by Rule 2.

Figure 3 presents examples of cross-spread of two virus strains, 1 and 2. The left panel shows the discriminant function, and the right panel displays the cross-spread of the two strains. The influence of the change in parameters (as an example, we only assume the *R*_0_ value of strain 1 changes) on the intersection of two strains is shown in the figure. In Figure 3A, there is no intersection. In Figure 3B,C, there are two intersections in the second and third domain, respectively. In Figure 3D, there is one intersection in the second domain. These intersection occurrences are in agreement with the aforementioned prediction rules.

In the first domain we have introduced *t*_a_ = *qm*_0_ − *t*_0_ as the initial time. To study the cross-spread in the time interval between *m* = 1 and *t*_a_ where *dF/dt* in *D*_21_ is difficult to be defined, one should use the *F*(*t*)-ladder method as follows:

The step size of CNIs of the first few steps in ladders 1 and 2 are given by *a*_1_, *b*_1_*a*_1_^2^, *b*_1_^3^*a*_1_^3^, *b*_1_^6^*a*_1_^4^, *b*_1_^10^*a*_1_^5^, *b*_1_^15^*a*_1_^6^, *b*_1_^21^*a*_1_^7^, *b*_1_^28^*a*_1_^8^, *b*_1_^36^*a*_1_^9^, etc., and *a*_2_, *b*_2_*a*_2_^2^, *b*_2_^3^*a*_2_^3^, *b*_2_^6^*a*_2_^4^, *b*_2_^10^*a*_2_^5^, *b*_2_^15^*a*_2_^6^, *b*_2_^21^*a*_2_^7^, *b*_2_^28^*a*_2_^8^, *b*_2_^36^*a*_2_^9^, etc., respectively (Equation (1)). Strain 1 spreads from the 1st step *a*_1_ at *t* = *c*_1_ − *d*_1_ on ladder 1 and strain 2 spreads from the 1st step *a*_2_ at *t* = *c*_2_ − *d*_2_ on ladder 2. The CNI and arrival time t of the two strains on their respective ladders are listed in Table 2. For example, let us take *c*_1_ = 7, *d*_1_ = 15, *c*_2_ = 5, *d*_2_ = 10. The earlier arrival times, in turn, are *t* = *c*_1_ − *d*_1_ = −8, *c*_2_ − *d*_2_ = −5, 2*c*_1_ − *d*_1_ = −1, 2*c*_2_ − *d*_2_ = 0, 3*c*_2_ − *d*_2_ = 5, 3*c*_1_ − *d*_1_ = 6, etc. By using Table 2, one can easily calculate the CNI at each arrival time as the parameters *a*_1_, *b*_1_, *a*_2_, *b*_2_ are given, compare the CNI values on two ladders and obtain the information on the cross-spread of two strains.

### 3.5. Examples of the Cross-Spread of Two Viruses

The epidemics that occurred in the UK from November 2020 to February 2022 provide a clear demonstration of the cross-spread phenomenon involving multiple viruses. This process can be divided into five stages, namely: (A) the alpha epidemic, (B) the delta invasion and cross-spread of two strains, (C) the delta dominant stage, (D) the omicron invasion and cross-spread of delta and omicron, and (E) the omicron dominant stage. In this section, we will specifically focus on studying the cross-spread of two strains during stages B and D.

Firstly, let us examine the cross-spread between the alpha strain (designated as strain (1) and the delta strain (designated as strain (2) during stage B. The spread of the alpha strain during stage A has been represented in Figure 1A, where we obtained the parameter *a*_1_ = *R*_0_^(1)^ = 3.9. The spread of the delta strain during stage C has been illustrated in Figure 4A, and we derived the parameter *a*_2_ = *R*_0_^(2)^ = 5.1 based on this data. By using *R*_0_^(1)^ and *R*_0_^(2)^ as inputs, we simulated the cross-spread of the two strains, as depicted in Figure 4B. Furthermore, utilizing all the parameters, *a_i_*, *b_i_*, *c_i_*, *d_i_* (*i* = 1,2), obtained from Figure 1A and Figure 4A,B, we constructed the discriminant function of the cross-spread and plotted the intersection of the two strains during stage B in Figure 4C. Interestingly, we discovered that one *t*_cr_ occurs at *t* = 71 in the region where *t* > *t*_m_ (*t*_m_ = 44.5), which is consistent with the prediction outlined in Rule 4.

Similarly, we investigated the cross-spread between the delta strain (strain 1) and the omicron strain (strain 2) during stage D. The results of this analysis are shown in Figure 5A–C. From Figure 5C, we observed that a *t*_cr_ appears at *t* = 42 in the region where *t* > *t*_m_ (*t*_m_ = 28.5), which is in agreement with the prediction made by Rule 4.

In both simulations, we assumed April 1 was the initial time for stage B, and November 11 was the initial time for stage D. Due to the uncertainty surrounding the exact timing of the delta invasion in stage B and the omicron invasion in stage D, we shifted the initial times *t* of stages B and D by several days. Remarkably, we found that the same values of *k*_1_, *k*_2_, *q*_1_, *q*_2_, *t*_0_^(1)^ were obtained, and only *t*_0_^(2)^ varied while still maintaining the invariant *t* + *t*_0_^(2)^.

## 4. Discussion

### 4.1. On the Simulation of COVID-19 Cases

The traditional SIR-type epidemic models depict the exponential growth of the number of infected individuals. However, empirical data have demonstrated that COVID-19 outbreaks do not exhibit exponential growth, but rather follow a three-parameter Gompertz growth function [13,14]. A new compartment model, known as the broken link model, has been proposed in the literature to explain the mechanism of Gompertz growth [15]. However, our proposed model is logically simple. In this model, we suggest that the spread of the virus depends on four parameters: *R*_0_, *k*, *q*, and *t*_0_. *R*_0_ describes the inherent infectious ability of the virus, *k* represents the strictness of social management, *q* represents the time needed for one step of infection, and *t*_0_ is an additional parameter that aligns with the starting date of the experimental data. By incorporating these four parameters, our four-parameter simulation accurately fits all existing COVID-19 epidemic data and will contribute to the prediction of future outbreaks. Moreover, the mechanism of Gompertz growth has been elucidated by our model.

### 4.2. On the Mutation of the SARS-CoV-2 Virus

The SARS-CoV-2 virus continuously undergoes mutations, giving rise to new strains. This ongoing mutation process is the reason why the pandemic has persisted for more than three years. As a result of natural selection, these new mutant strains possess higher infectious ability but lower lethality rates. Although the lethality rate of the new strains may be lower, it still results in a significant number of deaths. Therefore, since mutation occurrence is inevitable and costly, the key focus should be on reducing the epidemic probability of mutants. While humans cannot prevent the virus from mutating, they can hinder the mutant strains from dominating the competition. According to six prediction rules on the cross-spreading of two strains, it is highly unlikely for the intersection point *t*_cr_ to manifest under the following conditions: (1) when Δ > 0, there are three domains and in this case there will be no occurrence of *t*_cr_ within a domain if the period of this domain is short enough (i.e., *t*_s_ − *t*_a_ small, Δ small, or *t*_m_ large) or if the symbol of *D*_21_ within a domain does not align with the symbol of *F*_1_ − *F*_2_ at the initial time of this domain; (2) when Δ < 0, there is only one domain and in this case there will be no *t*_cr_ if the period of virus spread is short enough or if the symbol of *D*_21_ does not align with the symbol of *F*_1_ − *F*_2_ at the initial time.

An example to highlight the phenomenon of no occurrence of *t*_cr_ is the winter epidemic in China in 2022. Among tens of millions of SARS-CoV-2 cases in one city, no new mutant strain emerged or triumphed over the competition, with the exception of the original omicron strand. This suggests that the pandemic was effectively terminated within a short period, lasting only one month.

In summary, the strategies proposed by our model to control an epidemic have two main aspects. First, it is critical to prevent the occurrence of a second wave. Second, one should aim to avoid competition among mutant strains. In the latter scenario, minimizing the duration of virus spread is an efficient approach.

### 4.3. Dependence of Virus Infection Potential on Temperature and Humidity

The conformational equilibrium between the open and closed conformations of the receptor binding domain (RBD) of the spike (S) protein can be analyzed using first-principle techniques for a susceptible individual. The RBD can exist in either the open or closed position, known as the up or down conformation, respectively. The population of conformational states can be determined based on the free energy change during conformational transitions of the S protein. Let us denote the closed conformation as state A and the open conformation as state B. The Gibbs free energies of states A and B are represented by *G*_A_ and *G*_B_, respectively. Generally, if *G*_A_ is lower than *G*_B_, the RBD will primarily assume the inactive conformation A. Conversely, in order to initiate the infectious process, the equilibrium should shift towards the open conformation, implying that *G*_B_ should be lower than *G*_A_ [16]. The free energy *G* in a given conformation is related to the partition function *Z*, which can be expressed as follows:(12)$$G=-k_{B}TlnZ,Z=\sum_{n}e^{-\beta E_{n}}(\beta =\frac{1}{k_{B}T}).$$

Here *E_n_* represents the energy level of a given conformation,
(13)$${\left(E_{n}\right)}_{A,B}=V_{A,B}+(n+1/2)\hbar \omega_{A,B}$$

*V*_A,B_ represents two minima of conformational potential, respectively, and (*n* + 1/2)*ħω*_A,B_ represents the corresponding vibrational energy around the minimum of the conformational potential. By the summation of Boltzmann factor over vibration states one has
(14)$$\frac{Z_{A}}{Z_{B}}=e^{-\beta (V_{A}-V_{B})}Y_{A/B},Y_{A/B}=\frac{e^{\frac{1}{2}\beta \hbar \omega_{B}}-e^{-\frac{1}{2}\beta \hbar \omega_{B}}}{e^{\frac{1}{2}\beta \hbar \omega_{A}}-e^{-\frac{1}{2}\beta \hbar \omega_{A}}}$$

Let *T_C_* represent the phase transition temperature, which can be determined by Δ*G* = *G*_B_ − *G*_A_ = 0. From Equations (12) and (14) we obtain a simplified equation for *T_C_*
(15)$$\frac{2(V_{B}-V_{A})}{\hbar {(\omega }_{A}-\omega_{B})}=coth\frac{\hbar \omega_{A}}{2k_{B}T_{C}}(as\left|\frac{\omega_{A}-\omega_{B}}{\omega_{A}}\right|\ll 1)$$
and Δ*G* > 0 for *T* > *T_C_*, Δ*G* < 0 for *T* < *T_C_*. Therefore, if the environmental temperature decreases to *T* < *T_C_*, the conformational transition from the closed conformation to the open conformation occurs rapidly. This explains why the virus has higher transmission rates during the winter and the entrance to host cells is prioritized. Conversely, the summer season provides the most favorable conditions for virus elimination. In addition to temperature, the conformational equilibrium is influenced by humidity. The virus can be modeled as a charged sphere, and through the application of electrostatics principles to salty solutions, one can derive an expression for the potential at the surface of the charged sphere, where the dielectric constant *ε* is incorporated [17]. This implies that the elastic frequency *ω*^2^ should be replaced by *ω*^2^/*ε*. Considering water has a dielectric constant of *ε* = 80, the frequency parameter takes a reduced value of *ω*/9 in the presence of a fully salty solution rather than *ω* in a vacuum. Consequently, this provides a quantitative estimation of the virus’s infection potential, which is strongly dependent on humidity.

The above discussions on the susceptible individual can be extended to the population level, providing evidence that *R*_0_ is influenced by temperature and humidity. The point will be discussed in detail in the future work.

## 5. Conclusions

(1)We propose a logically simple model for analyzing the spread of the COVID-19 virus. The model is based on two fundamental assumptions: each viral strain possesses a basic reproduction number *R*_0_, which quantifies its transmission potential, and the instantaneous reproduction number (*R_t_* or *R_j_*) decreases gradually due to factors such as population immunity and interventions, which can be represented by an attenuation constant *k* (*k* < 1).(2)The daily number of infections (DNI) and the cumulative number of infections (CNI) versus time (*m*) are deduced based on the aforementioned two assumptions. By utilizing the explicit relation *m*(*t*) where *t* is the physical time, our simulations with the four parameters (*R*_0_, *k*, *q*, *t*_0_) demonstrate excellent agreement with all experimental data.(3)Insights obtained from typical plots of CNI vs. *m* (i.e., typical plots of *F*(*R*_0_, *k*; *m*)) provide valuable information regarding the conditions required for the decline of a viral infection wave, as well as an explanation for the occurrence of continuous second waves of infection and its preventive measures.(4)The persistence of the SARS-CoV-2 pandemic for more than three years can be attributed to frequent mutations. We thoroughly examine the cross-spread of two strains within a region and lay a theoretical foundation for designing strategies to avoid competition among mutant strains or hindering the mutant strains from dominating the competition.

## Figures and Tables

**Figure 1 viruses-15-01788-f001:**
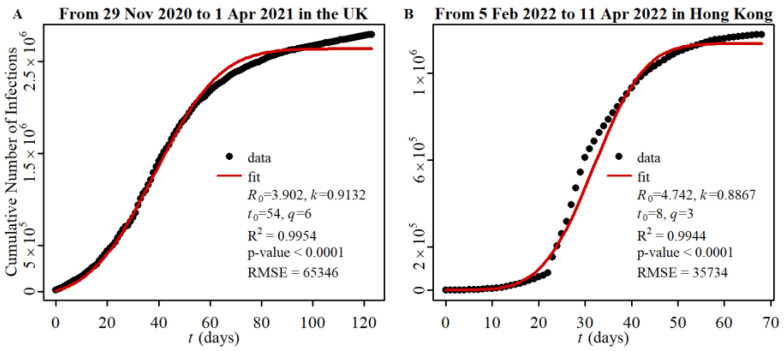
COVID-19 pandemics in the UK (**A**) and Hong Kong (**B**).

**Figure 2 viruses-15-01788-f002:**
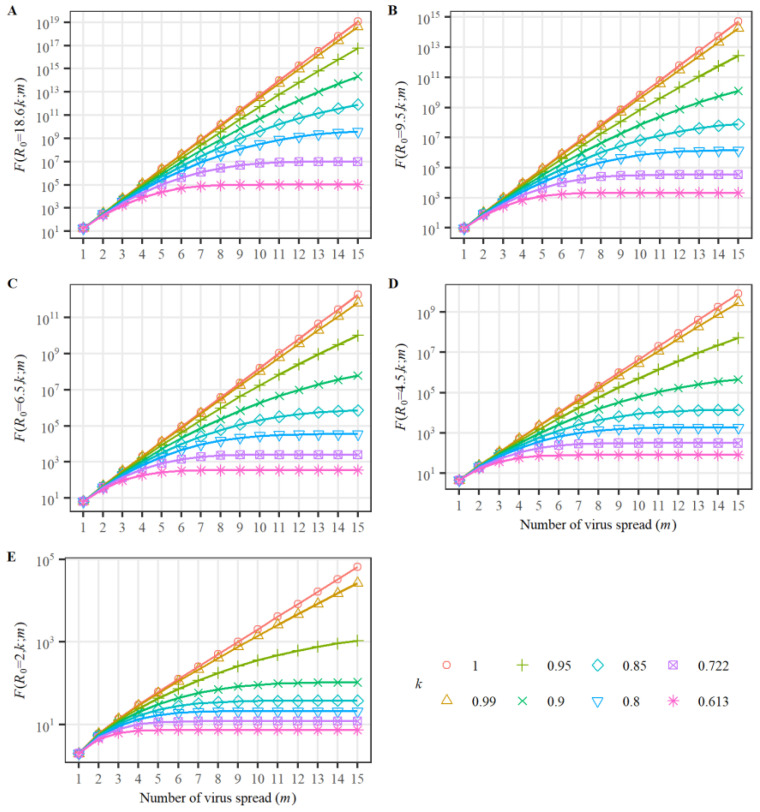
CNI functions *F*(*R*_0_, *k*; *m*) for several typical *R*_0_ values. (**A**): *R*_0_ = 18.6 (for Omicron BA.4, BA.5 in South Africa, 2022-1), (**B**): *R*_0_ = 9.5 (for Omicron B.1.1.529 in many countries, 2021-11), (**C**): *R*_0_ = 6.5 (for Delta in India, R0 = 5–8, 2020-10), (**D**): *R*_0_ = 4.5 (for Alpha in the UK, 2020-9, Beta in South Africa, 2020-5, Gamma in Brazil, 2020-11, *R*_0_ = 4–5), and (**E**): *R*_0_ = 2 (for SARS-CoV 2003, *R*_0_ = 2–3).

**Figure 3 viruses-15-01788-f003:**
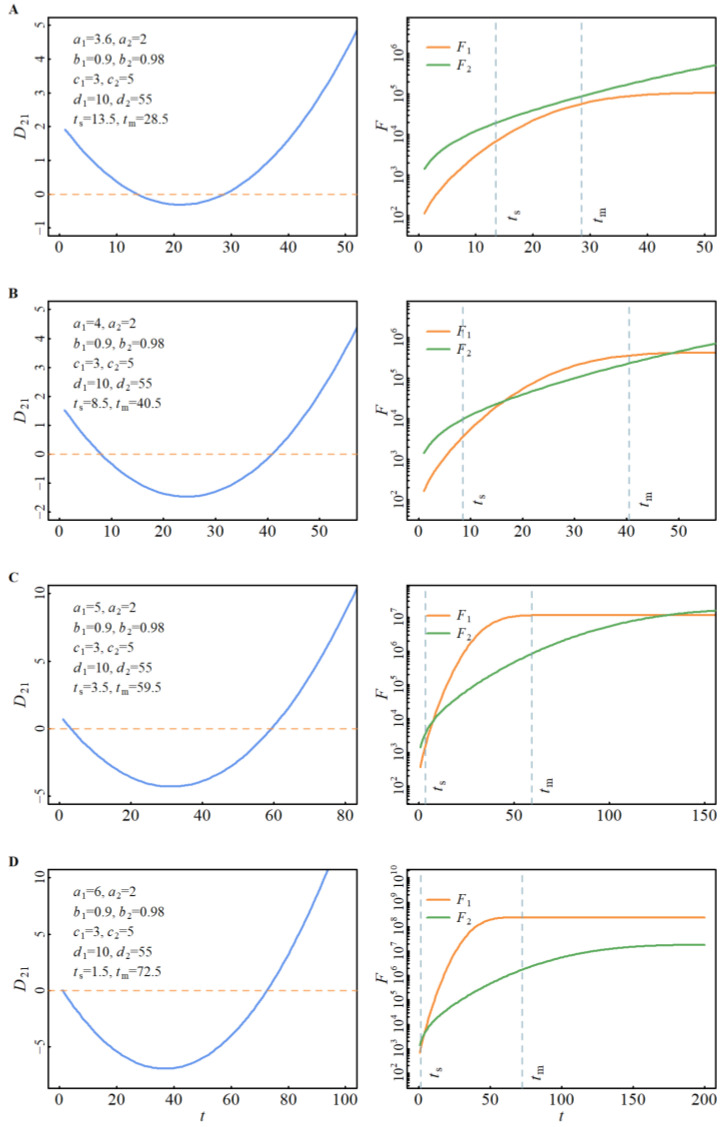
Discriminant Function and intersection occurrence.

**Figure 4 viruses-15-01788-f004:**
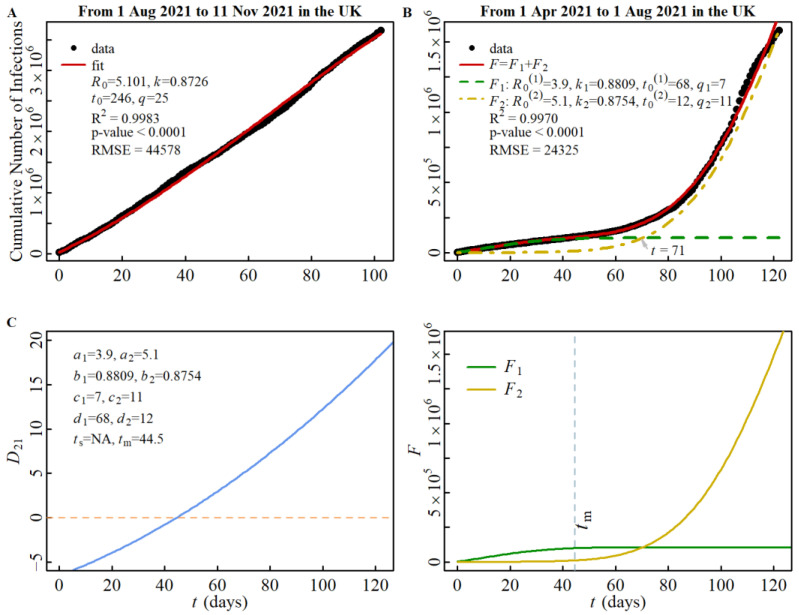
CNI simulation in the UK from November 2020 to November 2021. (**A**): CNI simulation of delta spread in stage C, (**B**): CNI simulation of alpha/delta spread in stage B, and (**C**): discriminant function and intersection of alpha/delta spread in stage B (left panel gives discriminant function and right panel the cross-spread of two strains). Note: CNI simulation of alpha spread in stage A from November 2020 to April 2021 has been plotted in Figure 1A.

**Figure 5 viruses-15-01788-f005:**
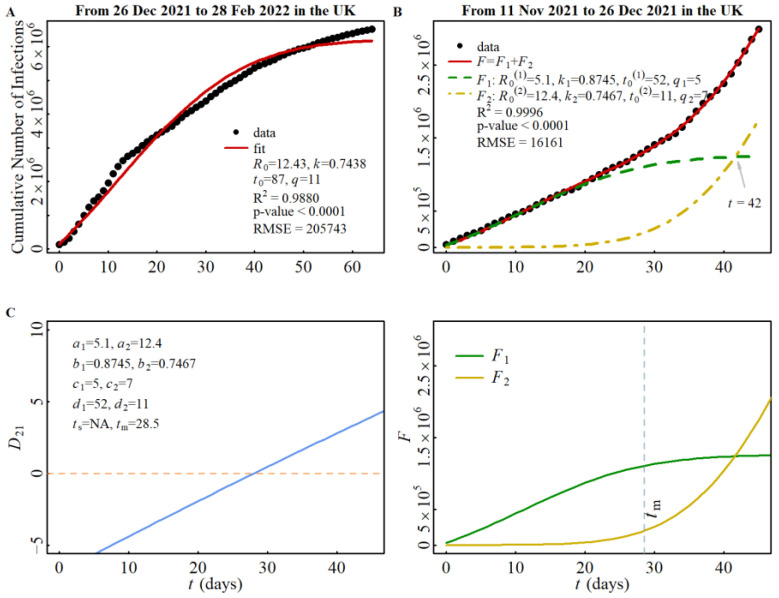
CNI simulation in the UK from November 2021 to February 2022. (**A**): CNI simulation of omicron spread in stage E (only the data of first omicron peak are used.); (**B**): CNI simulation of delta/omicron spread in stage D; and (**C**): Discriminant function and intersection of delta/omicron spread in stage D (left panel gives discriminant function and right panel the cross-spread of two strains).

**Table 1 viruses-15-01788-t001:** Parameters related to virus infection dying out.

*R* _0_	*k*_th_ (*m*_st_ = 15)	*E*_km_ = (1/*k*_th_)^7^	*N* _th_	*k*_cr_ (*N*_max_ = 10^7^)
18.6	0.613	30.7	10^5^	0.722
9.5	0.722	9.78	∼10^5^	0.8
6.5	0.75	7.49	∼10^4^	0.85
4.5	0.8	4.77	∼10^3^	0.9
2	0.9	2.09	∼10^2^	>0.95

**Table 2 viruses-15-01788-t002:** CNI and arrival time *t* of two strains on *F*(*t*)-ladder.

*t*	CNI
*c_i_ − d_i_* ^1^	*a_i_*
2*c_i_ − d_i_*	*a_i_ + b_i_a_i_* ^2^
3*c_i_ − d_i_*	*a_i_ + b_i_a_i_* ^2^ * + b_i_* ^3^ *a_i_* ^3^
4*c_i_ − d_i_*	*a_i_ + b_i_a_i_* ^2^ * + b_i_* ^3^ *a_i_* ^3^ * + b_i_* ^6^ *a_i_* ^4^
5*c_i_ − d_i_*	*a_i_ + b_i_a_i_* ^2^ * + b_i_* ^3^ *a_i_* ^3^ * + b_i_* ^6^ *a_i_* ^4^ * + b_i_* ^10^ *a_i_* ^5^
6*c_i_ − d_i_*	*a_i_ + b_i_a_i_* ^2^ * + b_i_* ^3^ *a_i_* ^3^ * + b_i_* ^6^ *a_i_* ^4^ * + b_i_* ^10^ *a_i_* ^5^ * + b_i_* ^15^ *a_i_* ^6^

^1^ *i* = 1, 2 refer to two strains, respectively, to save space only six steps of the *F*(*t*)-ladder are listed.

## Data Availability

Not applicable.

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
