# Peer review of "Mathematical Modelling of Virus Spreading in COVID-19"

_viruses, 2023, doi:10.3390/v15091788_

Round 1

Reviewer 1 Report (Previous Reviewer 2)

It is not clear to me how k can be assumed to be a constant. It seems to me that this might only be true for a very short time interval due to reasons which are also listed by the authors, namely interventions, mutations of the virus etc. It is also not clear how k can be determined. It is not clear to me how the model can be applied in real life situations.

The newly added references in line 43 of page 1 seem irrelevant and thus should be excluded. 

Line 97: j should be in lower index. 

The English has improved, but it needs further amelioration. Also, more care should be taken in using mathematical mode for formulas (and not using it when writing plain text).

Author Response

Reviewer 2 Report (Previous Reviewer 1)

The authors clear all the comments of the reviewers. This article in its present form is now in standard form for Acceptance and Publication. Therefore, I recommend it for Acceptance.

The authors clear all the comments of the reviewers. This article in its present form is now in standard form for Acceptance and Publication. Therefore, I recommend it for Acceptance.

Round 2

Reviewer 1 Report (Previous Reviewer 2)

I understand the method how k is calculated, my concern is about the concept of m. I'm not sure that the way it is introduced by the authors is realistic. Thinking about the epidemic as a branching process, various generations of different branches might exist at a same time. I don't think that "rounds of infection" exist in a way the authors introduce it. 

I can see that the model can give a good fit to actual data, but how well does it do in prediction? Also, it would be nice to see the fitting for a longer time period. E.g. in Fig.1., the two curves seem to start getting far from each other just before the end of the fitting period. 

I partly accept the authors' reply to my first question. It is true that some of the changes of the parameters are only significant in the long run. However, at the beginning of an epidemic, these changes might be much faster. 

As for the two references added: please explain why these two references were chosen. Why are they significant with respect to the paper's contents? One should be very careful to cite only papers which are relevant for the topic of the actual paper. Otherwise one risks going against publication ethics. 

The English has improved significantly. 

Author Response

This manuscript is a resubmission of an earlier submission. The following is a list of the peer review reports and author responses from that submission.

Round 1

Reviewer 1 Report

Review Report

Manuscript ID viruses-2528073

Mathematical Modelling of Virus Spreading in COVID-19

By:Liaofu Luo and Jun Lv

Reviewer comments

A MINOR revision is required to make the manuscript worth publishing. The

contents of the paper are good and contain new ideas. Anyhow, I would like to see

the following modifications, whether minor or major, in the revised version, which

would increase the strength of the paper and increase its potential readers, as well

as improve the current work.

 Abstract should be revised so that the findings of the paper may be clear

and attractive to the readers.

 The authors are requested to add more details regarding their original contributions in this manuscript.

 Update introductions with current research work.

 Please check the manuscript carefully for typos and grammar errors.

 Give parametric values in the paper for the consider model.

 Why the authors choose this method and what are the usefulness of this

method give more details.

 It is suitable that the authors cite some references related to their work for

the best presentation of the bibliography. For example,

Stability analysis and numerical solutions of fractional order HIV/AIDS

model. Stability analysis of fractional nabla difference COVID-19 model.

Computational and theoretical modeling of the transmission dynamics of

novel COVID-19 under Mittag-Leffler power law. Haar wavelet collocation

approach for the solution of fractional order COVID-19 model using Caputo

derivative. A fractional order Covid-19 epidemic model with Mittag-Leffler

kernel. Mathematical modeling of the COVID-19 epidemic with fear impact. A predatorprey model involving variable-order fractional differential

equations with Mittag-Leffler kernel.

 Future research direction must be shown in conclusion.

After revisions I recommend strongly this paper for publication.

Minor editing of English language required

Reviewer 2 Report

In this manuscript, the authors propose a mathematical model to analyse the spread of COVID-19. The parameters used in the model are the basic reproduction number and the attenuation constant k. The authors calculate the daily number of infections and the cumulative number of infections and show that these values are close to real world data. The model is used to explain the wave patter of daily new cases, to predict the occurrence of the second wave infectionand to understand the competitive spread of two viruses in a region. 

In my opinion, there are a number of issues which should be clarified. First, the meaning of k is not clear. What do you mean by "virus infections undergo a series of events in m steps that weaken with each step"? What kind of events? Do you have some biological medical references which support this assumption?

What do you mean by "For the virus to spread, it must continually re-infect the host population."? This sounds like the virus reinfects the whole population all the time. What does Rj mean? 

The definition of m is also dubious for me. I do not beleive that in all transmission chains, there are exactly the same number of transmissions.

The English of the manuscript is quite fine, a careful check could further improve the manuscript. One important thing, which is not about grammar but about layout: please use mathematical mode everywhere in formulas.